# Fabricating BiOCl Nanoflake/FeOCl Nanospindle Heterostructures for Efficient Visible-Light Photocatalysis

**DOI:** 10.3390/molecules28196949

**Published:** 2023-10-06

**Authors:** Heng Guo, Yangzhou Deng, Haoyong Yin, Juanjuan Liu, Shihui Zou

**Affiliations:** 1College of Materials & Environmental Engineering, Hangzhou Dianzi University, Hangzhou 310036, China; 212200116@hdu.edu.cn (H.G.); 221200047@hdu.edu.cn (Y.D.); yhy@hdu.edu.cn (H.Y.); 2Key Laboratory of Applied Chemistry of Zhejiang Province, Department of Chemistry, Zhejiang University, Hangzhou 310027, China

**Keywords:** FeOCl, BiOCl, heterostructures, photocatalysis, organic pollutants, nanospindles, nanoflakes

## Abstract

Fabricating heterostructures with abundant interfaces and delicate nanoarchitectures is an attractive approach for optimizing photocatalysts. Herein, we report the facile synthesis of BiOCl nanoflake/FeOCl nanospindle heterostructures through a solution chemistry method at room temperature. Characterizations, including XRD, SEM, TEM, EDS, and XPS, were employed to investigate the synthesized materials. The results demonstrate that the in situ reaction between the Bi precursors and the surface Cl^−^ of FeOCl enabled the bounded nucleation and growth of BiOCl on the surface of FeOCl nanospindles. Stable interfacial structures were established between BiOCl nanoflakes and FeOCl nanospindles using Cl^−^ as the bridge. Regulating the Bi-to-Fe ratios allowed for the optimization of the BiOCl/FeOCl interface, thereby facilitating the separation of photogenerated carriers and accelerating the photocatalytic degradation of RhB. The BiOCl/FeOCl heterostructures with an optimal composition of 15% BiOCl exhibited ~90 times higher visible-light photocatalytic activity than FeOCl. Based on an analysis of the band structures and reactive oxygen species, we propose an S-scheme mechanism to elucidate the significantly enhanced photocatalytic performance observed in the BiOCl/FeOCl heterostructures.

## 1. Introduction

Water pollution, particularly the discharge of toxic organic dyes by the textile industry, has become a notable environmental concern [1,2,3]. Rhodamine-B (RhB), a hazardous xanthene dye, has been identified as one of the most frequently encountered chemical compounds in liquid waste [4]. The presence of RhB waste poses significant risks to aquatic ecosystems. Additionally, exposure to this waste can be hazardous if ingested by humans or animals, leading to irritation of the skin, eyes, and respiratory tract [5]. Therefore, it is crucial to neutralize or decompose this dye waste before its final discharge into the environment. Traditional treatment methods such as coagulation, adsorption, and biological/electrochemical treatments have demonstrated a certain efficiency in removing RhB [5,6], but they often suffer from drawbacks such as a prolonged operation time and the generation of secondary sludge [7]. In recent years, semiconductor-based photocatalysis technology has garnered significant attention as a sustainable and efficient approach for degrading organic dyes [8,9,10,11,12]. Extensive research has been conducted on various photocatalysts, including oxides, oxyhalides, and polymer-based materials [13,14,15]. Compared to traditional methods, semiconductor-based photocatalysis offers several advantages. It operates under mild conditions and utilizes solar energy as a sustainable power source. Moreover, the effective conversion of organic pollutants into harmless byproducts also reduces the need for sludge disposal [7]. Nevertheless, the low efficiency of photocatalysts, caused by the low light utilization and rapid recombination of photogenerated electron–hole pairs, presents a significant challenge for practical implementation [16].

Engineering heterojunctions in photocatalysts has become a recognized strategy for enhancing photocatalytic efficiency. This approach offers advantages such as efficient light harvesting and the spatial separation of photogenerated electron–hole pairs [16]. Over the past few years, substantial progress has been achieved in developing highly efficient heterojunction photocatalysts. Furthermore, several systematic reviews have been published regarding heterojunction photocatalysts [16,17,18,19]. However, the fabrication of heterostructures with rich interfaces and intricate nanoarchitectures in a facile, cost-effective, and controllable manner still presents a significant challenge [20].

FeOCl, a material with a van-der-Waals-layered structure, has emerged as a functional material for many applications [21,22,23]. In particular, the weak interlayer interactions make it an excellent host for intercalation reactions [24,25], while the reducible electronic properties render it an exceptional catalyst in heterogeneous Fenton reactions [26,27]. As a semiconductor with small bandgap (~1.8 eV), FeOCl can be activated by visible-light irradiation [26]. However, the rapid recombination of photogenerated carriers hampers its efficacy as a visible-light photocatalyst. One potential solution to this problem is the construction of FeOCl-based heterostructures with other suitable semiconductors [28,29,30,31]. For instance, Song et al. [32] synthesized an AgSCN/AgCl/FeOCl nanosheet heterojunction with a novel interface structure, which exhibited 40.7 and 4.1 times higher photocatalytic and photo-Fenton catalytic activities than FeOCl, respectively. Vadivel et al. reported that FeOCl/g-C_3_N_5_ [30] and FeOCl/Bi_5_O_7_Br heterostructures [33] are excellent photocatalysts for the degradation of organic pollutants. Zhao et al. [29] reported that Z-scheme g-C_3_N_4_/FeOCl heterostructures efficiently accelerate charge transfer and promote photocatalytic activity. According to the literature, nanoarchitectures [34] and interfacial structures [15,35] profoundly influence the photocatalytic performance of heterostructures. One-dimensional/two-dimensional heterojunction photocatalysts with diverse interfacial contact forms possess unique advantages for photocatalysts [36]. However, the fabrication of FeOCl-based heterostructures commonly involves the calcination of a mixture of FeCl_3_ and pre-synthesized semiconductors (e.g., g-C_3_N_4_ [29], TiO_2_ [37], and WS_2_ [38]). Due to the complicated interplay between reactants, it is difficult to control the morphology of FeOCl. In most cases, the resulting FeOCl exhibits 2D nanosheet structures [39]. The development of FeOCl-based heterostructures with intricate nanoarchitectures (1D/2D) and adjustable interfacial structures remains an open research area.

Herein, we report the facile synthesis of novel BiOCl nanoflake/FeOCl nanospindle heterostructures. BiOCl was selected as the semiconductor to form a heterostructure with FeOCl due to its lattice and energy-level compatibility [35]. Moreover, it shares a similar layer structure with FeOCl [24], and its nucleation and growth behavior can be easily controlled by adjusting the solution composition (ethylene glycol/water ratio) [40]. Using FeOCl nanospindles as the start materials and regulating their in situ reaction with Bi precursors in ethylene glycol solution enabled the bounded nucleation and growth of BiOCl on the surface of the FeOCl nanospindles. Stable interfacial structures were formed between BiOCl nanoflakes and FeOCl nanospindles by using Cl^−^ as the bridge. Regulating the Bi-to-Fe ratios can optimize the interfacial structures of BiOCl/FeOCl, thereby enhancing the separation of photogenerated carriers and accelerating the photocatalytic degradation of RhB. The optimized heterostructures containing 15% BiOCl exhibited 90-fold higher visible-light photocatalytic activity than FeOCl. The rate constant value of 15% BiOCl/FeOCl (0.055 min^−1^) exceeded that of the reported FeOCl-based photocatalysts (in the absence of H_2_O_2_) [31,37] and was even comparable to some FeOCl-based photo-Fenton systems [28,41].

## 2. Results and Discussion

The synthetic procedures for FeOCl and BiOCl/FeOCl are schematically illustrated in Figure 1a. First, FeOCl was synthesized by calcinating FeCl_3_·6H_2_O at 180 °C for 1.5 h. The scanning electron microscopy (SEM) image in Figure 1b reveals that the obtained sample featured nanospindle structures, with lengths of 100–300 nm and diameters of ~30 nm. The X-ray diffraction (XRD) patterns indicated that all the diffraction peaks can be attributed to orthorhombic FeOCl [27] (PDF#24-1005), thus confirming the successful obtainment of pure FeOCl through the low-temperature calcination of FeCl_3_·6H_2_O (Figure 2).

The BiOCl/FeOCl heterostructures were synthesized using a solution chemistry method at room temperature. Specifically, a certain amount of Bi(NO_3_)_3_·5H_2_O was dissolved in ethylene glycol and subsequently introduced into the FeOCl suspension. The resulting mixture was vigorously stirred at room temperature for 3 h, facilitating a cation-exchange reaction between the Bi^3+^ ions and FeOCl to form BiOCl. The XRD patterns of BiOCl/FeOCl (Figure 2) exhibited characteristic peaks of tetragonal BiOCl [40] (PDF#85-0861), confirming that Bi^3+^ can react with FeOCl to produce BiOCl. The diffraction peaks of BiOCl were well resolved, suggesting that the as-produced BiOCl had a high crystallinity. Furthermore, no discernible peak shifts were observed in the XRD patterns of FeOCl, indicating that the growth of BiOCl took place on the surface of FeOCl rather than doping within the FeOCl lattice [42].

Figure 1c–g present the SEM images of BiOCl/FeOCl with varying Bi-to-Fe ratios. The SEM image of 5% BiOCl/FeOCl (Figure 1c) revealed that the nanospindle structure of FeOCl was maintained, although the surface became rough, indicating the uniform anchoring of BiOCl onto the surface of the FeOCl nanospindles. When the Bi-to-Fe ratio increased to 7%, the anchored BiOCl grew and partially transformed into nanoflakes (Figure 1d). Further increasing the Bi-to-Fe ratio led to the generation of more BiOCl nanoflakes on the surface of the FeOCl nanospindles (Figure 1e–g). Elemental mapping (Figure 3) demonstrated a uniform distribution of the Fe, O, and Bi elements throughout the sample, indicating the close contact between the BiOCl nanoflakes and the FeOCl nanospindles.

The morphology evolution in the BiOCl/FeOCl heterostructures with varying Bi-to-Fe ratios was also confirmed using transmission electron microscopy (TEM). As shown in Figure 4, pure FeOCl featured a nanospindle morphology. When the Bi-to-Fe feed ratio increased from 5% to 20%, a gradual decrease in nanospindles and a corresponding increase in nanoflakes were observed, indicating the progressive transformation of FeOCl into BiOCl. It is worth noting that, in all the BiOCl/FeOCl samples, close contact between the FeOCl nanospindles and BiOCl was observed. The resultant heterostructures possessed abundant BiOCl/FeOCl interfaces, which could facilitate charge transfer and promote photocatalysis performance.

Based on the above experimental results and the relevant literature [40,43], we proposed the following mechanism for the formation of BiOCl/FeOCl heterostructures. First, Bi(NO_3_)_3_·5H_2_O was dissolved in ethylene glycol to form a bismuth alkoxide solution [44]. When introduced to the FeOCl suspension, the bismuth alkoxide underwent hydrolysis, resulting in the formation of (BiO)^+^ ions [40]. Since the insoluble FeOCl was the only source of Cl^−^, the (BiO)^+^ ions would react with FeOCl in situ to form BiOCl. By using Cl^−^ as a bridge, the newly formed BiOCl would be firmly attached to the FeOCl surface [43]. Regulating the Bi-to-Fe ratio can control the nucleation and growth rate of BiOCl, which consequently determines the morphology of BiOCl/FeOCl. When the Bi-to-Fe ratio was very low (e.g., 5%), the reaction between the (BiO)^+^ ions and Cl^−^ mainly took place on the surface of FeOCl. Due to the high solubility of bismuth alkoxide in ethylene glycol, the as-generated (BiO)^+^ ions were highly dispersed in the FeOCl suspension. The high accessibility of the FeOCl surface to (BiO)^+^ facilitated the nucleation of BiOCl, resulting in a uniform anchoring of BiOCl on the surface of the FeOCl nanospindles. Consequently, 5% BiOCl/FeOCl exhibited a similar nanospindle structure with a rough surface. As the Bi-to-Fe ratio increased, the surface Cl^−^ of FeOCl was progressively consumed. The additional (BiO)+ ions continued to react with the bulk FeOCl, resulting in the growth of BiOCl nuclei into nanoflakes. Notably, increasing the Bi-to-Fe ratio significantly increased the number of BiOCl nanoflakes, but barely changed the thickness of the BiOCl nanoflakes. This is likely due to the same solvent composition used in the solution chemistry method. According to our previous study [40], the thickness of BiOCl nanoflakes is closely related to the ethylene-glycol-to-water (EG-W) ratio because it determines the hydrolysis rate of bismuth alkoxide and the growth rate of BiOCl. In the present study, the EG-W ratio was kept constant at 5/25 and the thickness of the BiOCl nanoflakes was ~20 nm, which was in good agreement with our previous study [40].

The surface composition and chemical states of FeOCl and 15% BiOCl/FeOCl were investigated using X-ray photoelectron spectroscopy (XPS). As shown in Figure 5a, the Fe 2*p* spectra of FeOCl can be deconvoluted into three sets of peaks (711.5/725.3, 714.1/727.9, and 719.5/733.7 eV) corresponding to Fe^II^, Fe^III^, and the satellite peak, respectively. This observation is consistent with previous studies [27,45], indicating the presence of a typical mixed-valence iron oxide and chloride. Interestingly, when FeOCl reacted with Bi^3+^ to form the 15% BiOCl/FeOCl heterostructures, a noticeable downshift in the Fe 2*p* peaks was observed. These results, on one hand, confirm the successful transformation of FeOCl into BiOCl; on the other hand, they indicate the existence of an electronic interaction between FeOCl and the as-formed BiOCl. Specifically, the formation of the –O–Fe–Cl–Bi–O– interface weakened the chemical bond strength of Fe–Cl and led to electron redistribution [43]. Notably, both 15% BiOCl/FeOCl and BiOCl exhibited sharp Cl 2*p* peaks at 198.1 and 199.7 eV, whereas FeOCl exhibited broad Cl 2*p* peaks at 198.6 and 200.2 eV (Figure 5b). These results suggest that the reaction between Bi^3+^ and FeOCl converted a significant portion of the surface FeOCl into BiOCl. This hypothesis is further supported by the similar Bi4*f* peaks observed in 15% BiOCl/FeOCl and BiOCl (Figure 5d). In addition, 15% BiOCl/FeOCl showed a weaker O1*s* signal from adsorbed water molecules compared to FeOCl (Figure 5c). This weakening was attributed to the transformation of FeOCl to BiOCl, which increased the lattice oxygen (Bi-O) [43]. The shift in the O1*s* peaks, similar to the shifts observed in the Cl2*p* and Fe2*p* peaks, confirmed the interfacial electronic interaction between BiOCl and FeOCl. According to our previous studies [15,46], the interfacial electronic interaction ensures bounded nucleation and growth, leading to the formation of heterostructures with rich interfaces. Herein, the interfacial electronic interaction between FeOCl and Bi^3+^ ensured the bounded nucleation and growth of BiOCl on the surface of FeOCl, as evidenced by the nanospindle structure of 5% BiOCl/FeOCl. With an increase in the Bi-to-Fe ratio, the anchored BiOCl further grew into nanoflakes, resulting in BiOCl/FeOCl heterostructures with rich interfaces.

The superior photocatalytic performance of heterostructures, in comparison to their individual components, is generally related to two primary factors: enhanced light harvesting and promoted electron–hole separation. Herein, the optical properties of FeOCl, 15% BiOCl/FeOCl, and BiOCl were investigated using UV–vis diffuse reflectance spectra. As shown in Figure 6a, 15% BiOCl/FeOCl exhibited absorption responses similar to FeOCl, suggesting that the formation of BiOCl nanoflakes on the surface had a negligible impact on the light responses of FeOCl. The bandgap energies (E_g_) of the catalysts were determined using Tauc plots:αh*ν* = A(h*ν* − E_g_)^2^
where α, h, *ν*, E_g_, and A are the absorption coefficient, Planck constant, light frequency, bandgap energy, and proportionality constant, respectively. As shown in Figure 6b, the bandgap energies were 3.18, 1.70, and 1.75 eV for BiOCl, FeOCl, and 15% BiOCl/FeOCl, respectively. It is noteworthy that the bandgap of 15% BiOCl/FeOCl was only slightly larger than that of pristine FeOCl, further confirming that the presence of BiOCl nanoflakes on the surface had minimal impact on the light-harvesting capabilities of FeOCl. The energy band positions of BiOCl and FeOCl were determined using the Mott–Schottky method. As shown in Figure 6c,d, pristine BiOCl and FeOCl exhibited positive slopes in the Mott–Schottky curves, indicating that they are both n-type semiconductors. For n-type semiconductors, the flat band potential (V_fb_) is usually close to the conduction band (E_CB_). In Figure 6c,d, the flat band potentials (V_fb_) of BiOCl and FeOCl were determined to be −1.02 and 0.78 eV. Consequently, the E_CB_ of BiOCl and FeOCl was −1.02 and 0.78 eV, respectively. By using the equation E_g_ = E_VB_ − E_CB_, the valence band (E_VB_) of BiOCl and FeOCl was calculated to be 2.16 and 2.48 eV, respectively. The band structures of FeOCl and BiOCl are depicted by the inset figure of Figure 6c,d.

The photocurrents were measured under visible-light irradiation (λ ≥ 420 nm, 100 mW cm^−2^) to investigate the separation of electron–hole pairs. As shown in Figure 7, the photocurrent density of 15% BiOCl/FeOCl was twice that of FeOCl, implying a lower recombination rate of photo-generated electron–hole pairs. This lower recombination rate potentially contributed to the enhanced catalytic activity [46]. In addition, the photocurrent response of 15% BiOCl/FeOCl exhibited a minimal decrease over time, indicating a good stability of 15% BiOCl/FeOCl heterostructures.

The photocatalytic activities of the BiOCl/FeOCl catalysts were evaluated by studying the degradation of RhB under visible-light irradiation. As shown in Figure 8a, FeOCl degraded only 3.6% of the RhB within 60 min, which was likely due to its fast recombination rate of photogenerated electron–hole pairs. Interestingly, once BiOCl was anchored on the surface of the FeOCl nanospindles to form BiOCl/FeOCl heterostructures, the photocatalytic activity was boosted. In particular, 15% BiOCl/FeOCl degraded 91.3% of the RhB within 40 min, which is significantly higher than that of FeOCl. Because 15% BiOCl/FeOCl exhibited similar light responses to FeOCl (Figure 6a), its superior activity in comparison with FeOCl should originate from the promoted separation and transfer of photogenerated carriers (Figure 7). It is important to note that the physical mixture of FeOCl and BiOCl (15% BiOCl/FeOCl-mix) exhibited much lower photocatalytic activity (11% degradation within 60 min) than the 15% BiOCl/FeOCl heterostructures (96.2% degradation within 60 min). Their distinct photocatalytic performance was likely due to their different interface structures [15]. The BiOCl/FeOCl heterostructures constructed using the solution chemistry method had rich interfaces because of the bounded nucleation and growth (i.e., nanoscale hybridization) of BiOCl on the surface of FeOCl. These heterointerfaces promoted the separation and transfer of photogenerated electron–hole pairs, which consequently led to superior photocatalytic activity [15,35]. The ratio of Bi to Fe is an important parameter for BiOCl/FeOCl heterostructures because it can control the growth of BiOCl to regulate the heterointerface. Specifically, when the Bi-to-Fe ratio is below 15%, increasing the Bi-to-Fe ratio will promote the growth of BiOCl nanoflakes (Figure 1) and increase the BiOCl/FeOCl interfaces. As a result, the photocatalytic performance of BiOCl/FeOCl improves with an increase in the Bi-to-Fe ratio (Figure 8a). However, once the Bi-to-Fe ratio exceeds 15%, most of the surface FeOCl is converted into BiOCl (Figure 5b). Further increasing the Bi-to-Fe ratio no longer leads to additional improvement in the BiOCl/FeOCl interfaces. The excess BiOCl might block the surface and influence the charge/mass transfer. To this end, 20% BiOCl/FeOCl showed similar photocatalytic activity to 15% BiOCl/FeOCl. It is worth highlighting that the majority of the dyes were removed by photocatalytic degradation under visible-light irradiation rather than being adsorbed onto 15% BiOCl/FeOCl (Figure 8d). Moreover, 15% BiOCl/FeOCl exhibited a promising efficacy in the visible-light-induced photocatalytic degradation of methylene blue (Figure 8d), showcasing its potential for the elimination of various other dye compounds.

To better quantify the photocatalytic performance of these catalysts, we fitted the photodegradation behavior using a pseudo-first-order kinetics model (Figure 8b) and calculated the rate constant (*k*). As shown in Figure 8c, the *k* value of *X*% BiOCl/FeOCl (*X* = 5, 7, 10, 15, 20) catalysts were 10–90 times higher than that of FeOCl. Besides, the rate constant of 15% BiOCl/FeOCl was ~30 times higher than that of 15% BiOCl/FeOCl-mix. These results confirm that the construction of BiOCl/FeOCl heterostructures can greatly promote the photocatalytic degradation of RhB under visible-light irradiation. The rate constant followed the trend of FeOCl < 5% BiOCl/FeOCl < 7% BiOCl/FeOCl < 10% BiOCl/FeOCl < 15% BiOCl/FeOCl > 20% BiOCl/FeOCl, highlighting the importance of heterointerfaces in photocatalysis [15,47]. Notably, the rate constant value of 15% BiOCl/FeOCl (0.055 min^−1^) exceeded that of most reported FeOCl-based photocatalysts (in the absence of H_2_O_2_) [31,37] and was even comparable to some FeOCl-based photo-Fenton systems [28,41]. These results, together with the facile synthesis, clearly demonstrate the superiority of the BiOCl nanoflake/FeOCl nanospindle heterostructures.

The recycling stability of 15% BiOCl/FeOCl was assessed by conducting four consecutive cycles of visible-light photodegradation. As shown in Figure 8e, the photodegradation behavior in the fourth cycle closely resembled that of the first cycle, indicating the good reusability and stability of 15% BiOCl/FeOCl. Furthermore, the spent catalyst (15% BiOCl/FeOCl-spent) exhibited a similar XRD pattern (Figure 2) and SEM and TEM images (Figure 9) to that of the fresh catalyst. These findings provide additional confirmation of the good recycling stability of 15% BiOCl/FeOCl.

To gain deep insight into the reaction mechanism, we also conducted radical-trapping experiments to identify the reactive oxygen species (ROS) involved in the photocatalysis. Disodium ethylenediaminetetraacetate (Na_2_-EDTA), p-benzoquinone, and tert-butanol were used as scavengers to trap h^+^, O_2_^•−^, and ^•^OH, respectively [35]. As displayed in Figure 8f, adding tert-butanol into the reaction system barely changed the degradation rate, excluding the involvement of ^•^OH in the photocatalytic degradation of RhB. In stark contrast, the addition of p-benzoquinone remarkably decreased the degradation efficiency from 96.2% to 20.4% within 60 min, indicating O_2_^•−^ as the key ROS involved in the photocatalytic degradation of RhB over BiOCl/FeOCl. Similarly, the decrease in the degradation efficiency induced by the addition of Na_2_-EDTA suggests that h^+^ is engaged in the photocatalysis. It is important to note that the photogenerated electrons of FeOCl cannot react with O_2_ to produce O_2_^•−^ because the conduction band of FeOCl (0.78 V vs. NHE) is much more positive than the potential of O_2_/O_2_^•−^ (−0.33 V vs. NHE, Figure 10). The production of reactive O_2_^•−^ should, therefore, originate from the reaction of O_2_ with the photogenerated electrons of BiOCl, whose CB potential (−1.02 V vs. NHE) is more negative than E_0_(O_2_/O_2_^•−^). On the other hand, because the VB potential of FeOCl (2.48 V vs. NHE) is more positive than that of BiOCl (2.16 V vs. NHE), the photogenerated holes of FeOCl would be more reactive towards the oxidation of RhB. These results suggest BiOCl and FeOCl as oxidation and reduction photocatalysts, respectively. Considering that BiOCl and FeOCl are n-type semiconductors with staggered band structures, an S-scheme heterojunction can be formed when they are hybridized on a nanoscale [18]. Owing to the band bending and the built-in electric field, the photogenerated electrons in the conduction band (CB) of FeOCl will combine with the photogenerated holes in the valence band (VB) of BiOCl. The photogenerated electrons in the CB of BiOCl and the holes in the VB of FeOCl are preserved to engage in the photocatalytic reaction. In summary, the S-scheme BiOCl/FeOCl heterojunction collectively achieved charge separation, thereby boosting the photocatalytic performance. Tailoring the Bi-to-Fe ratios to increase the BiOCl/FeOCl interfaces would promote photogenerated electron transfer from FeOCl to BiOCl. As a result, more photogenerated electrons and holes can be preserved in the CB of BiOCl and the VB of FeOCl, respectively, to produce O_2_^•−^ and h^+^ for RhB degradation [18].

## 3. Materials and Methods

### 3.1. Synthesis of BiOCl/FeOCl

FeOCl nanospindles were synthesized through the calcination of FeCl_3_·6H_2_O (≥99%, Aladdin Chemicals, Shanghai, China). Notably, the calcination conditions played a crucial role in determining the structure of FeOCl. In this study, FeOCl was synthesized through the calcination of FeCl_3_·6H_2_O at 180 °C for 1.5 h (ramp rate: 5 °C min^−1^). The optimized conditions allowed for the synthesis of pure FeOCl nanospindles.

BiOCl/FeOCl heterostructures were synthesized using a solution chemistry method at room temperature (25 °C). Taking 15% BiOCl/FeOCl (the molar ratio of Bi to Fe is 15%) as an example, 0.15 g of the obtained FeOCl nanospindles were dispersed in 25 mL of H_2_O with ultrasound for 15 min. Subsequently, 5 mL of ethylene glycol containing 0.102 g of Bi(NO_3_)_3_·5H_2_O (≥99%, Aladdin Chemicals, Shanghai, China) was added into the above suspension and vigorously stirred for 3 h. The solid products were collected using centrifugation, washed with distilled water several times, and dried at 80 °C overnight. BiOCl/FeOCl with other Bi-to-Fe ratios were synthesized using the same method.

### 3.2. Characterization

The crystalline structures of the samples were measured using X-ray diffraction (XRD, Rigaku Ultimate IV diffractometer using Cu Kα radiation, Tokyo, Japan). The morphologies of the samples were characterized using scanning electron microscopy (SEM, Hitachi SU8100, Tokyo, Japan) and transmission electron microscopy (TEM, JEOL JEM-1230, Tokyo, Japan). The UV–Vis diffuse reflectance spectra were recorded on a Shimadzu UV-2450 spectrophotometer (Shimadzu, Kyoto, Japan). The X-ray photoelectron spectra (XPS) were recorded on a Thermo Scientific Escalab 250 spectrometer (Waltham, MA, USA). All the binding energies were referenced to the C1*s* peak of contaminant carbon at 284.8 eV. Electrochemical measurements (zero-biased photocurrent and Mott–Schottky plots) were carried out on a CHI 660E electrochemical workstation (CHI instruments, Shanghai, China) with a 0.1 M Na_2_SO_4_ solution as the electrolyte. The experimental setup involved a standard three-electrode system comprising a Pt counter electrode, an Ag/AgCl reference electrode, and an indium tin oxide (ITO) electrode modified with the catalyst as the working electrode. A 300 W Xe lamp with cutoff filters (λ ≥ 420 nm, CEL-HXF300, Beijing China Education AuLight Technology Co., Ltd., Beijing, China) was utilized as a visible-light source. The Mott–Schottky plots were recorded at a frequency of 2 kHz. The measured potentials versus Ag/AgCl were converted to the normal hydrogen electrode (NHE) scale using the equation E(vs. NHE) = E(vs. Ag/AgCl) + 0.197.

### 3.3. Catalytic Test

In brief, 50 mg of catalysts and 100 mL of rhodamine B (RhB, 15 mg L^−1^) were vigorously stirred for 60 min in the dark to achieve an adsorption–desorption equilibrium. Subsequently, the suspension was subjected to visible-light irradiation using a high-pressure Xe lamp with an optical filter (300 W, λ ≥ 420 nm, Beijing China Education AuLight Technology Co., Ltd., Beijing, China) as the light source. At a given time interval, 2 mL of the suspension was extracted and centrifuged. The liquid was analyzed using a Shimadzu UV-2450 spectrophotometer at 553 nm to determine the concentration of RhB. Radical-trapping experiments were conducted under similar conditions except for the addition of scavengers (10 mM) into the suspension prior to the visible-light irradiation. Disodium ethylenediaminetetraacetate (Na_2_-EDTA), tert-butanol, and p-benzoquinone were used as the scavengers to trap the h^+^, •OH, and O_2_^•−^ radicals, respectively.

## 4. Conclusions

BiOCl nanoflake/FeOCl nanospindle heterostructures with tunable interfacial structures were successfully fabricated using a solution chemistry method at room temperature. Owing to the promoted separation of electron–hole pairs and the enhanced generation of ROS (O_2_^•−^ and h^+^), the optimized heterostructures containing 15% BiOCl exhibited a 90 times higher rate constant than FeOCl in the visible-light-driven photodegradation of RhB. These findings verify that the fabrication of heterostructures with abundant interfaces and intricate nanoarchitectures is an effective strategy for optimizing photocatalytic activity.

## Figures and Tables

**Figure 1 molecules-28-06949-f001:**
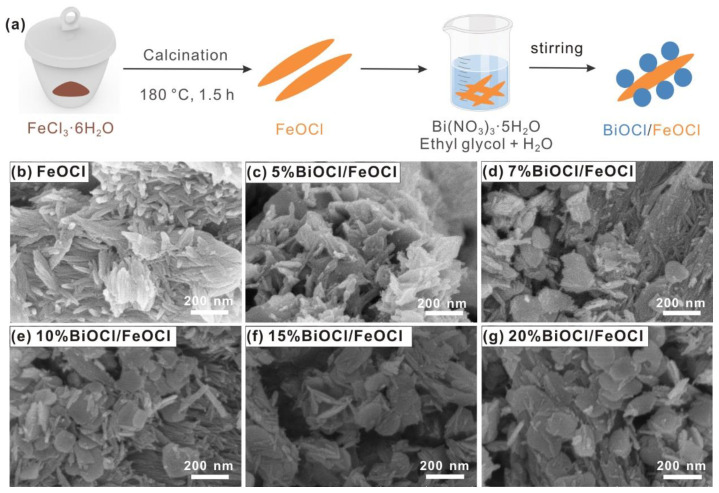
(**a**) Schematic illustration of the synthesis process of BiOCl/FeOCl; (**b**–**g**) SEM images of FeOCl and X% BiOCl/FeOCl (*X* = 5, 7, 10, 15, 20).

**Figure 2 molecules-28-06949-f002:**
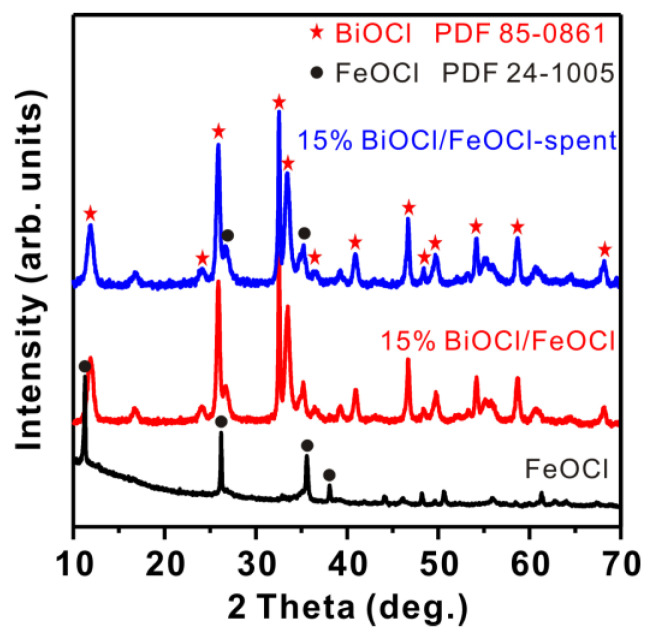
XRD patterns of FeOCl, 15% BiOCl/FeOCl, and 15% BiOCl/FeOCl-spent.

**Figure 3 molecules-28-06949-f003:**
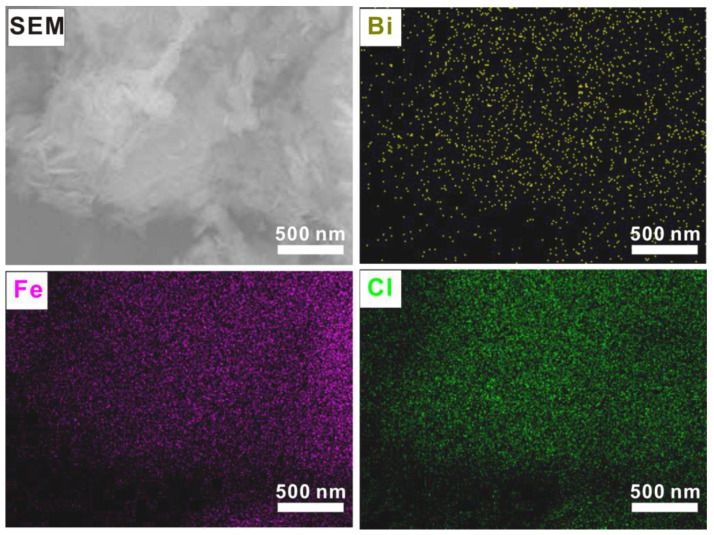
Elemental mapping of 15% BiOCl/FeOCl.

**Figure 4 molecules-28-06949-f004:**
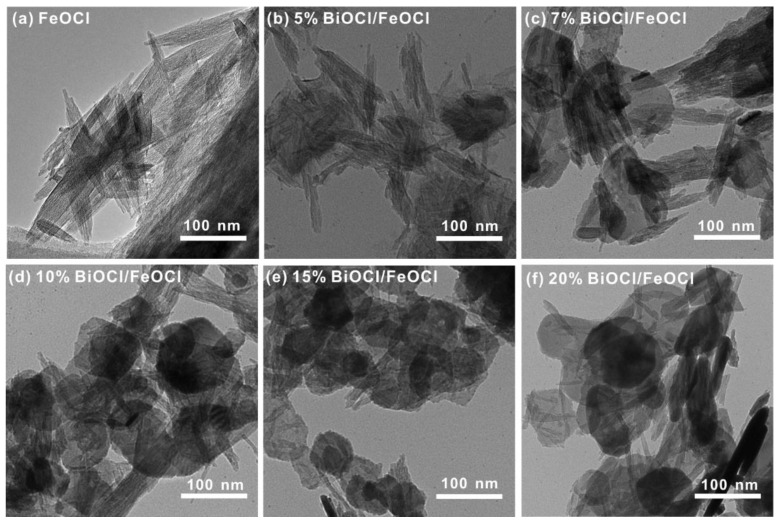
TEM images of (**a**) FeOCl and (**b**–**f**) *X*% BiOCl/FeOCl (*X* = 5, 7, 10, 15, 20).

**Figure 5 molecules-28-06949-f005:**
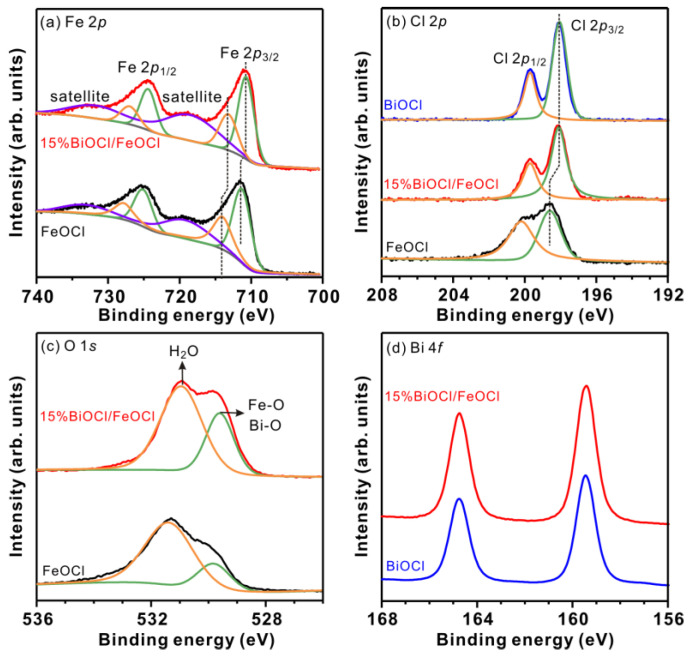
XPS spectra and fitting curves of FeOCl, 15% BiOCl/FeOCl, and BiOCl. (**a**) Fe 2*p*; (**b**) Cl 2*p*; (**c**) O 1*s*; and (**d**) Bi 4*f*.

**Figure 6 molecules-28-06949-f006:**
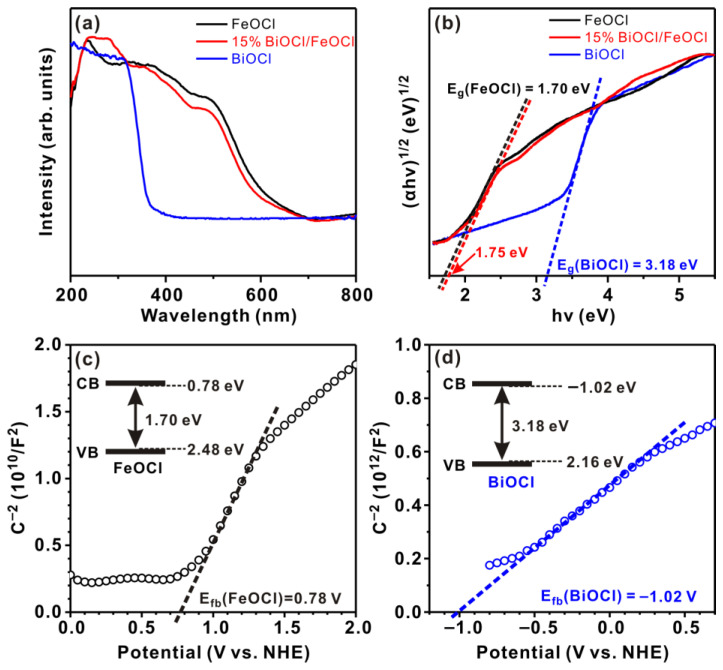
(**a**) UV–vis diffuse reflectance spectra and (**b**) the corresponding Tauc curves of FeOCl, 15% BiOCl/FeOCl, and BiOCl; Mott–Schottky plots of (**c**) FeOCl and (**d**) BiOCl.

**Figure 7 molecules-28-06949-f007:**
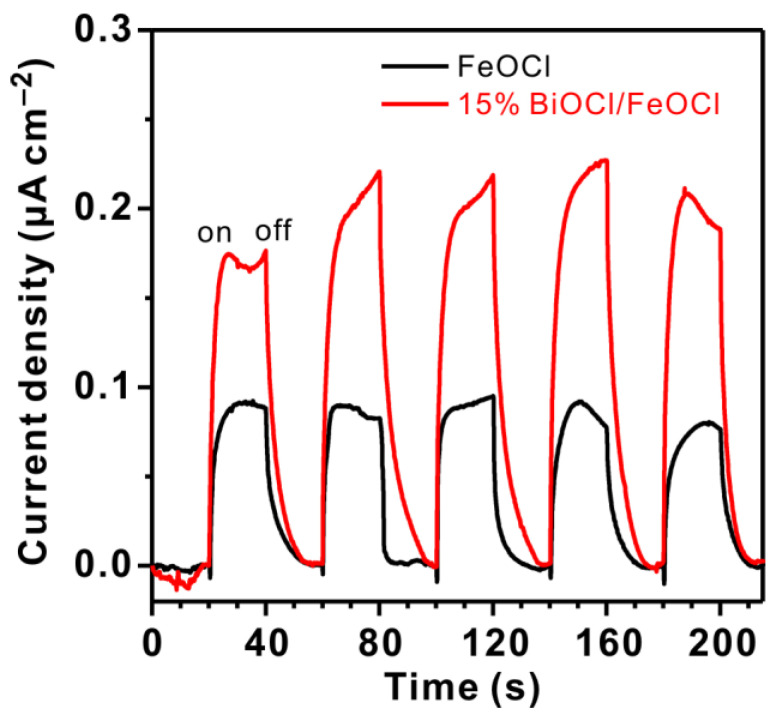
Photocurrent responses of 15% BiOCl/FeOCl and FeOCl under visible-light irradiation (λ ≥ 420 nm).

**Figure 8 molecules-28-06949-f008:**
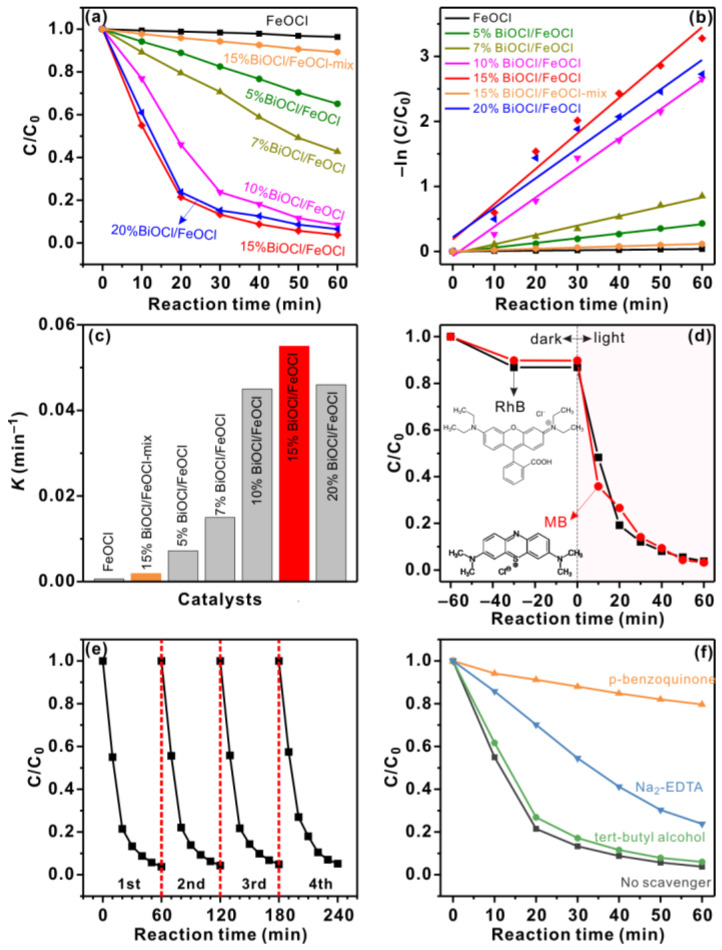
(**a**) Photocatalytic degradation of RhB under visible-light irradiation over FeOCl and *X*% BiOCl/FeOCl (*X* = 5, 7, 10, 15, 20); (**b**) kinetic linear simulation curves and (**c**) corresponding rate constants; (**d**) photocatalytic degradation of methylene blue (MB) and RhB under visible-light irradiation over 15% BiOCl/FeOCl; (**e**) cyclic experiments of 15% BiOCl/FeOCl; and (**f**) degradation of RhB in the presence of different scavengers on 15% BiOCl/FeOCl. Reaction conditions: 25 °C, 50 mg catalyst, 100 mL RhB or MB (15 mg L^−1^), and high-pressure Xe lamp (300 W) with an optical filter (λ ≥ 420 nm).

**Figure 9 molecules-28-06949-f009:**
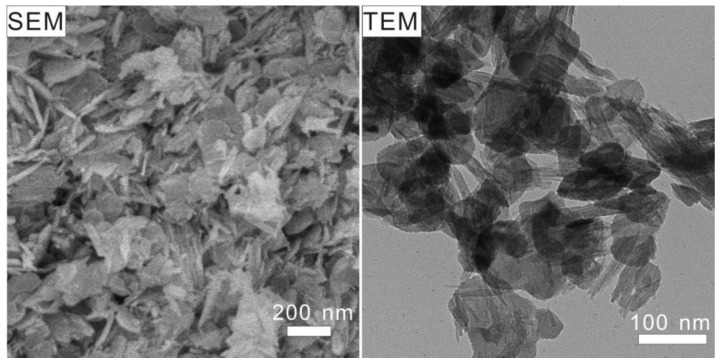
SEM and TEM images of 15% BiOCl/FeOCl-spent.

**Figure 10 molecules-28-06949-f010:**
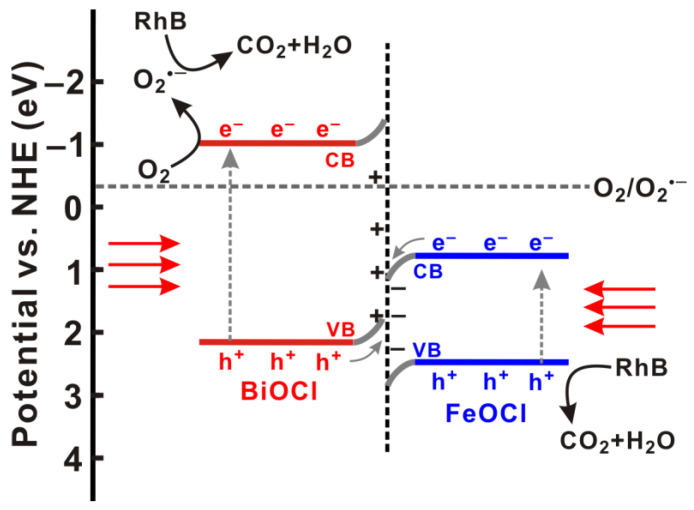
Schematic illustration of photocatalytic degradation of RhB over BiOCl/FeOCl.

## Data Availability

The data presented in this study are available from the corresponding author upon request.

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
