# Peer review of "Fabricating BiOCl Nanoflake/FeOCl Nanospindle Heterostructures for Efficient Visible-Light Photocatalysis"

_molecules, 2023, doi:10.3390/molecules28196949_

Round 1
Reviewer 1 Report
In this manuscript, the authors prepared BiOCl/FeOCl composite by using a a solution chemistry method for the degradation of Rhodamine B under visible light irradiation. The result showed that photocatalytic capability of 15% BiOCl /FeOCl was 30 folds higher than 15% BiOCl/FeOCl-mix. The work may be accepted after major modification.
1. The English language of the manuscript is poor. Please revise the whole manuscript by asking a native speaker.
2. The novelty and purpose of the paper are too vague. By strengthening the literature review, authors can make a compelling case that the novel aspect of the work is relevant.
3. What is the superiority of your prepared materials in comparison with other materials? This should be explained in the introduction section
4. The introduction should include a literature summary about removing Rhodamine B from aqueous solutions.
5. The comparative investigations with previously published works about the removal efficiency of Rhb from other research groups should be presented and discussed with many details
6. Please provide more photocatalytic degradation experiments for the removal of other organics.
7. Some basic characterizations of BiOCl/FeOCl composite photocatalyst should be provided before and after reaction, such as XRD, SEM and XPS.
8. How about the mineralization situation of Rhb? The authors should provide the details of TOC.
9. The electron paramagnetic resonance (EPR) spectroscopy results should been given to prove hydroxyl radicals.
10. A regeneration test should be provided in the paper.
11. The mechanism analysis must be performed based on the investigation of energy band structure of samples
The English language of the manuscript is poor. Please revise the whole manuscript by asking a native speaker.
Author Response
We sincerely thank all reviewers for their valuable feedback that we have used to improve the quality of our manuscript. Please see our point-by-point response in the attachment.

Reviewer 2 Report
In this work, BiOCl nanoflakes/FeOCl nanospindles heterostructures were synthesized, characterized, and used for photodegradation of RhB. The paper is very well-written, and I think it can be considered for publication after minor revisions. Please see the comments.
- How were calcination time and temperature chosen and what is their effect on composite photocatalytic performance?
- The removal within the first 60 min should be shown as illustrated in Figure 7. Is it an effective adsorbent too?
- The experiment conditions should be added to the figure captions
- Is the photocatalyst reusable?
Author Response

(The authors gave the same response as above.)

Reviewer 3 Report
The present paper focuses on the development of the new type photocatalyst based on the heterojunction between BiOCl and FeOCl and its application into the dye-degradation. Especially, the authors performed the various characterizations including XRD, SEM, TEM, EDS, and photocurrent measurements. Thus, the submitted article can be helpful for the researchers working in this field. But, I have some concerns and comments as follows.
i) In the “Materials and Methods” part, the author used the Hg lamp for the catalytic test. As I know, the spectrum of Hg lamp is focused on the UV or near-UV region relative to that in Xe lamp. Why did they use the Hg lamp? Is there any special reason?
ii) In 100 lines of Page 4, the author argued that “It is worth noting that in all BiOCl/FeOCl samples, FeOCl nanospindles are in close contact with BiOCl nanoflakes.”. As I checked the Figure 4b, it was hard to see the close contact between BiOCl and FeOCl.
iii) The analyses of XPS spectra look not clear to me. One of the key observation in the XPS measurement is that the 15% BiOCl/FeOCl heterostructure shows the changes of major peaks in the Fe2p spectra. To do that, the author should do the analysis of XPS spectra based on the fitting with the sum of Gaussians as they did in Figure 5b and 5c. From this way, they should determine the degree of the energy change in terms of value.
iv) In the optical property, the author argued that “As shown in Fig. 6a, 15% BiOCl/FeOCl exhibits similar absorption responses as FeOCl …”. In my view, the UV-visible spectrum of 15% BiOCl/FeOCl heterostructure shows a slight difference to that of FeOCl. I recommend that the author should estimate the optical bandgap based on the Kubelka–Munk equation. Such an approach enables the authors to check the underlying optical property in the as-synthesized material.
v) In the “Introduction” part, the authors introduced the photocatalyst based on the heterostructures around 40 lines. There are many different types of the heterostructure-based photocatalysts such as polymer heterostructures, core/shell nanostructures, or multi-shell structure. But, the previous studies and the literatures were not properly introduced. I recommend that such explanations should be added in the introduction part and the following reference should be properly cited.
- J. Low et. al., Adv. Mat., 2017, 29, 1601694
- Kumar et. al., J. Photochem. Photobiol. A 2023, 439, 114591
- Liu et. al., Chem. Eng. J. 2020, 393, 124719
- Singh et. al., React. Chem. Eng. 2023, 8, 1072-1082
There are some minor comments;
In abstract, they should fix as follows.
“… is an appealing strategy for the optimization of photocatalyst.”
“with Bi3+ to form 15% BiOCl/FeOCl heterostructures”
Especially, the author used the words of “15%BiOCl/FeOCl” and “15% BiOCl/FeOCl” in the main text. I suggest that the authors use the correct word of “15% BiOCl/FeOCl” for consistency. In addition, the author should carefully check the main text and improve the English language for the potential readers.
There are some minor comments;
In abstract, they should fix as follows.
“… is an appealing strategy for the optimization of photocatalyst.”
“with Bi3+ to form 15% BiOCl/FeOCl heterostructures”
Especially, the author used the words of “15%BiOCl/FeOCl” and “15% BiOCl/FeOCl” in the main text. I suggest that the authors use the correct word of “15% BiOCl/FeOCl” for consistency. In addition, the author should carefully check the main text and improve the English language for the potential readers.
Author Response

(The authors gave the same response as above.)

Reviewer 4 Report
Manuscript ID:
Title: Fabricating BiOCl Nanoflakes/FeOCl Nanospindles Heterostructures for Efficient Visible-light Photocatalysis
Well focused of this work is facile way prepare the material, characterization, and its’s degradation study towards rhodamine B. Manuscript explored the details about synthesis and characterization of the material. The author has written the manuscript in brief and well-organized form. Besides this, there are still some concerns/negligence which need to be addressed before its acceptance to publishable. I hope that the mentioned points will be resolved well. Please see the comments below thoroughly.
Recommendation: Revision
Comments:
1. As author mentioned the statement about production of more h+ and O2 •− by 2D/1D heterointerface. I suggest the author not to let your prior statement as such and give emphasis behind contribution for more production of more h+ and O2 •− by 2D/1D heterointerface.
2. Why do authors choose BiOCl Nanoflakes/FeOCl Nanospindles Heterostructure catalysts?
3. Rational design of Fe based catalyst can be elaborated in introduction section with following reference:doi.org/10.1016/j.cej.2021.129312
4. The author has claimed S-scheme type of the material. In my knowledge, to be S-scheme type in heterojunction, material should be composed of n-type semiconductors. However, the way of writing makes the statement controversial. Please elaborate the logic in a more systematic way (from line 222-233).
5. Keep full stop after number of references (line 25-26, 28-29, and so on.). Also, give space before the reference (line 35, 36, 40 ….). Please correct these issues throughout the manuscript.
6. How is your degradation phenomenon different from other degradation techniques such as plasma treatment, DBD, chemical and electrochemical degradation? There are other techniques and materials/catalyst synthesized for the study of dyes degradation. To insight/elaborate the degradation phenomena, you can prefer the following literatures as follows.
Chemosphere, Volume 309, Part 1, December 2022, 136638; http://dx.doi.org/10.1016/j.chemosphere.2022.136638,Catalysts 2020, 10(5), 546; https://doi.org/10.3390/catal10050546
Author Response

(The authors gave the same response as above.)

Round 2
Reviewer 1 Report
Ultimately, the authors answered all the questions perfectly, so the current form can be accepted.
Reviewer 3 Report
The authors tried to improve the quality of the manuscript and fully addressed the issues raised from the reviewer.
I recommend the publication of the current manuscript as it is.
Reviewer 4 Report
Accept